META-RESEARCH ARTICLE

# Shortcomings of reusing species interaction networks created by different sets of researchers

**Chris Brimacombe** [1]*, **Korryn Bodner**[2], **Matthew Michalska-Smith**[3,4], **Timothée Poisot**[5,6], **Marie-Josée Fortin**[1]

**1** Ecology and Evolutionary Biology, University of Toronto, Toronto, Ontario, Canada, **2** MAP Centre for Urban Health Solutions, St. Michael's Hospital, Unity Health Toronto, Toronto, Ontario, Canada, **3** Department of Ecology, Evolution and Behavior, University of Minnesota, Minneapolis, Minnesota, United States of America, **4** Department of Plant Pathology, University of Minnesota, Minneapolis, Minnesota, United States of America, **5** Département de sciences biologiques, Université de Montréal, Montréal, Québec, Canada, **6** Centre de la Science de la Biodiversité du Québec, Montréal, Québec, Canada

* Chris.Brimacombe@mail.utoronto.ca

## Abstract

Given the requisite cost associated with observing species interactions, ecologists often reuse species interaction networks created by different sets of researchers to test their hypotheses regarding how ecological processes drive network topology. Yet, topological properties identified across these networks may not be sufficiently attributable to ecological processes alone as often assumed. Instead, much of the totality of topological differences between networks—topological heterogeneity—could be due to variations in research designs and approaches that different researchers use to create each species interaction network. To evaluate the degree to which this topological heterogeneity is present in available ecological networks, we first compared the amount of topological heterogeneity across 723 species interaction networks created by different sets of researchers with the amount quantified from non-ecological networks known to be constructed following more consistent approaches. Then, to further test whether the topological heterogeneity was due to differences in study designs, and not only to inherent variation within ecological networks, we compared the amount of topological heterogeneity between species interaction networks created by the same sets of researchers (i.e., networks from the same publication) with the amount quantified between networks that were each from a unique publication source. We found that species interaction networks are highly topologically heterogeneous: while species interaction networks from the same publication are much more topologically similar to each other than interaction networks that are from a unique publication, they still show at least twice as much heterogeneity as any category of non-ecological networks that we tested. Altogether, our findings suggest that extra care is necessary to effectively analyze species interaction networks created by different researchers, perhaps by controlling for the publication source of each network.

**Data Availability Statement:** All data and code to reproduce our results are available from OSF (https://osf.io/my9tv/).

**Funding:** CB was supported by an Ontario Graduate Scholarship (https://osap.gov.on.ca/OSAPPortal/en/A-ZListofAid/PRDR020870.html). M-JF acknowledges the support of the Natural Sciences and Engineering Research Council of Canada Discovery Grant (no. 5134; www.nserc-crsng.gc.ca/index_eng.asp) and the Canada Research Chair in Spatial Ecology. The funders had no role in study design, data collection and analysis, decision to publish, or preparation of the manuscript.

**Competing interests:** The authors have declared that no competing interests exist.

## Introduction

Network approaches are routinely used as a tool to analyze ecological systems [1–4]. This popularity extends to species interaction networks which model ecological communities, where nodes represent species and edges represent their corresponding species interactions [5]. Due to the effort needed to observe species and their interactions in situ, creating species interaction networks requires a tremendous amount of resources for their adequate construction [6–8]. Given this requisite effort, instead of creating their own networks, ecologists often reuse available species interaction networks which happen to be created by different sets of researchers, to test their own ecological hypotheses [9,10]. These available species interaction networks have therefore been used in many ecological studies including when determining the complexity of networks [11], identifying common topological properties across networks [12,13], and evaluating how network topology is shaped by species traits [14], environmental factors/space [7,15–17], and time [18–20].

A current crux of reusing available species interaction networks created by different sets of researchers, however, is the unwanted topological differences that can exist due to the lack of consistency in the way ecological systems are translated into networks by different sets of researchers [17,21–27] (Fig 1). While some criticisms related to this issue had been raised in the 1980s/90s, e.g., Paine (1988) [28] and Polis (1991) [29], with the increasing availability to the internet and growth in computational power, a renewed interest in networks was sparked, and many of these concerns were overlooked [30].

**Fig 1. Potential sources of topological heterogeneity that influence researchers' interpretation of a plant–pollinator community as a bipartite network.** Here, the observed plant–pollinator community (green oval) is translated into a researcher's network representation (thought bubble). Sources of topological heterogeneity between different researchers' network interpretations of a community could be introduced from: (i) observing different biological and environmental drivers (purple text) that influence the community's interactions, (ii) the different selected sampling strategies (orange text) that influence which biological and environmental factors are included during a researcher's observation, and (iii) the different selected network construction methods (blue text) researchers use to design a species interaction network.

**Table 1. Classes of topological heterogeneity that influence species interaction networks, some sources of this topological heterogeneity, a description of the source, and references.**

| Classes of heterogeneity | Source | Description of source | Example references |
|---|---|---|---|
| Biological and environmental drivers | Type of environment | Abiotic conditions influence topology (e.g., plant–pollinator networks are structured differently across a temperature gradient) | Welti and Joern (2015) [31], Pellissier and colleagues (2018) [7] |
| | Population sizes | Species abundances influence probability of interaction (e.g., topological differences in communities with high abundances vs. low abundances of all species) | Vázquez and colleagues (2007) [32], Vázquez and colleagues (2009) [33] |
| | Interaction frequencies | Number of interaction events an organism has influences the probability interactions are recorded in a network (e.g., topological differences in networks when cryptic interactions are included) | Pringle and Hutchinson (2020) [30], Pringle (2020) [34] |
| Sampling strategies | Temporal elements of study | The duration and time interval of observation used to characterize a community influence its representation as a network (e.g., topological differences when networks are constructed from data collected over a day vs. week) | CaraDonna and Waser (2020) [18], Schwarz and colleagues (2020) [19], CaraDonna and colleagues (2021) [20] |
| | Spatial elements of study | The extent and area resolution of observation used to characterize a community influence its representation as a network (e.g., topological differences when networks are constructed from data collected in a patch vs. a forest) | Galiana and colleagues (2018) [35], Galiana and colleagues (2022) [36] |
| Network construction methods | Selection of interaction types | Interaction types differently influence communities (e.g., topological differences between mutualistic and antagonistic systems) | Thébault and Fontaine (2010) [37], Allesina and Tang (2012) [38] |
| | Node resolution | Organismal classification and targeted species influences topology (e.g., topological differences when nodes represent species vs. genus or when nodes additionally represent ontogentic stages) | Hemprich-Bennett and colleagues (2021) [39], Bodner and colleagues (2022) [40] |

In order to effectively reflect on these criticisms and overall problems that occur when reusing species interaction networks created by different sets of researchers, we thought it necessary to first have a vocabulary to do so. As such, we introduce a framework that partitions how the totality of these topological differences—topological heterogeneity—between species interaction networks created by different sets of researchers can originate, by broadly organizing its sources into 3 classes (Table 1): biological and environmental drivers, sampling strategies, and network construction methods. The biological and environmental drivers class consists of sources of topological heterogeneity that arise from the different (a)biotic conditions that shape species and their interactions across different communities. For example, abiotic conditions including temperature can influence whether a species persists as well as modify their interactions [31]. Likewise, biotic drivers such as population sizes can influence both the existence and strength of interactions [32,33]. The sampling strategies class consists of sources of topological heterogeneity that arise from the different study design decisions made by researchers when observing species and their interactions and determines which effects from (a)biotic factors are included in the network, e.g., how larger sampling area and larger sampling time can capture greater environmental factors. The network construction methods class consists of sources of topological heterogeneity that are introduced via the different decisions made by researchers when constructing each network, e.g., only using plant species from a single genus (Fig 2A) or including unidentified species in the network (Fig 2B). In combination, these classes of topological heterogeneity make it incredibly difficult to decipher which topological properties in species interaction networks might be due to the ecological process of interest rather than due to unwanted sources of heterogeneity.

Structural differences between species interaction networks are not problematic per se since topological heterogeneity is necessary for determining drivers of that topology. However, large amounts of topological heterogeneity between networks created by different sets of researchers may be indicative of networks that lack commensurability [43]. While some

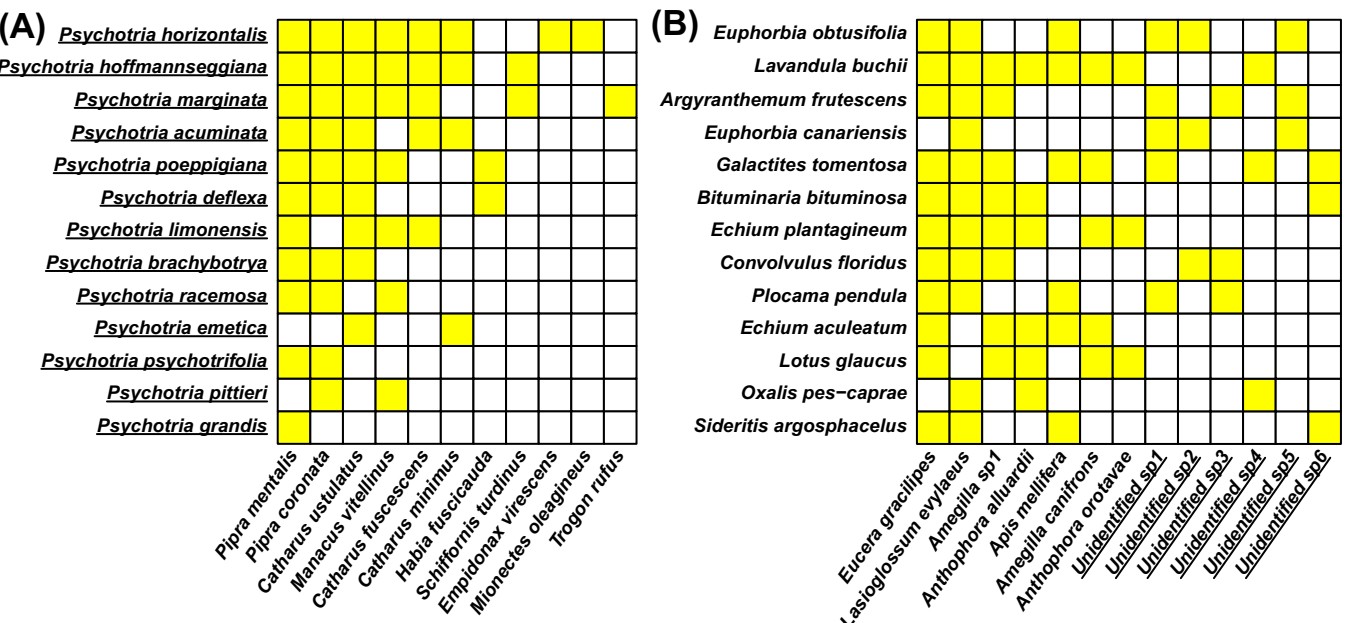

**Fig 2. Matrix representations of 2 bipartite species interaction networks from www.web-of-life.es; an open species interaction network database.** Yellow boxes in each matrix indicate the presence of an interaction between species at the corresponding row (plants) and column (animals). **(A)** Seed-dispersal network from Poulin and colleagues (1999) [41], where all plant species (underlined) are from the genus *Psychotria*. **(B)** Subset of the plant–pollinator network from Stald (2003) [42], which includes a large number of unidentified pollinator species (underlined; 34 of the 54 total pollinator species (not all shown here) in the whole network).

studies attempt to control for inconsistencies in the way species interaction networks are created by different researchers, e.g., controlling for sampling effort [44] or network size [45], these controls do not account for all associated unwanted topological heterogeneity when using the many different topological metrics adopted by ecologists. Hence, evaluating the amount of topological heterogeneity present in species interaction networks created by different researchers is a necessity. One approach to do this is by comparing the dispersion of network topology within species interaction networks to other real world networks that are not significantly hampered by the classes of heterogeneity listed in Table 1. If a system is accurately portrayed by its own networks, we would expect these networks to have a small amount of dispersion expressed within the metrics used to capture their topology.

As an attempt to quantify ecological topological heterogeneity, we used the largest set of bipartite networks and measured the amount of topological dispersion in (i) species interaction networks compared to non-ecological networks; and (ii) species interaction networks created by the same set of researchers (i.e., networks from the same publication) compared to the species interaction networks each a product of their own publication. We quantified differences in network topology using directed graphlet correlation distance [46], a heuristic method that measures the Euclidean distance between networks, where networks closer together are those that are more topologically similar.

To measure topological heterogeneity, we evaluated the total dispersion in directed graphlet correlation distances between networks of the same domain (defined below). As most ecological networks do not have metadata regarding the conditions under which each network was created (i.e., their associated biological and environmental drivers, sampling strategies, and network construction methods), we could not partition topological heterogeneity across the heterogeneity classes. However, as we quantified the total dispersion in non-ecological

networks from different domains which are, to a large extent, not hampered by the 3 classes of heterogeneity, we could use these dispersion values to estimate the total amount of topological heterogeneity as a result of all 3 heterogeneity classes in species interaction networks. Furthermore, we compared the total dispersion of species interaction networks from the same publication to those that are not from the same publication to determine if there are topological biases due to the ways in which different sets of researchers construct networks.

## Materials and methods

### Data

A total of 3,476 bipartite networks were used in this study (see Table 2 for a description of all networks and their domains). Of the non-ecological bipartite networks included in our analysis, 1,830 were of the crime domain, 109 were of the journal domain, 245 were of the legislature domain, 172 were of the actor domain, 194 were of the sports domain, and 203 were of the microbiome domain. We classified microbiome networks as non-ecological since, among other properties, they were not built using observational data (instead, for example, by swabs and subsequent RNA sequencing) and had concrete definitions for their edges/nodes (i.e., locations on the human body where a bacterial operational taxonomic unit was found)—2 stark features that differ from species interaction networks (our ecological networks). See Aagaard and colleagues (2013) [47] for a thorough description of how patients were selected, and operational taxonomic units were sampled, which were the data we used to build the microbiome networks. Although microbiome networks could be considered ecological, we believed that their topological heterogeneity would more resemble non-ecological networks and thus grouped them accordingly. Except for sports networks that we constructed for this paper and whose data were obtained from Lahman (2021) [48], www.basketball-reference.com, and www.hockey-reference.com, all non-ecological networks were obtained from Michalska-Smith and Allesina (2019) [13]. Of the 723 species interaction networks used in this

**Table 2. Description of bipartite networks used in this study.**

| Network domain | Node set 1 | Node set 2 | A connection forms when | Subgroup each network represents |
|---|---|---|---|---|
| Species interaction | Species | Species | A species feeds on another | Ant–plant, host–parasite, plant–herbivore, plant–pollinator, or seed–dispersal community |
| Actor | Actors | Film | An actor appears in a film | Action, adventure, animation, comedy, crime, documentary, drama, family, fantasy, foreign, history, horror, music, mystery, romance, science fiction, thriller, tv movie, war, or western movie genre across a number of years |
| Crime | Type of crimes | A city's neighborhoods | A crime occurs in a city's neighborhood | Chicago, Denver, Minneapolis, San Francisco, or Washington of a given day in 2016 |
| Journal | Authors | Academic journals | An author publishes in an academic journal | All authors that published in one of The American Naturalist, Ecography, Ecological Application, Ecological Monographs, Ecology, Ecology Letters, Journal of Animal Ecology, Journal of Applied Ecology, Journal of Ecology, or Oikos, and all other academic journals within a given year (2006–2016) |
| Legislature | Legislators | Bills | A legislator votes positively for a bill | The US House, the US Senate, the UN General Assembly, or the European Parliament of a given year (1941–2016) |
| Microbiome | Bacteria operational taxonomic unit (OTU) | Locations on a human patient | A bacteria OTU is found on patient's site | A single patient |
| Sports | Athletes | Teams | An athlete plays for a team | The NBA, NHL, or MLB, of a given season (1950–2020) |

All bipartite networks were connected and had at least 5 nodes in either disjoint sets of nodes.

analysis (obtained from Brimacombe and colleagues (2022) [10]), 10 were ant–plant networks, 97 were host–parasite networks, 41 were plant–herbivore networks, 298 were plant–pollinator networks, and 277 were seed–dispersal networks. All networks that were included in our analysis were unweighted (i.e., interactions between nodes were binary).

We included non-ecological systems for comparison in our study given the strict definitions used to define their systems, thereby eliminating much of the biological and environmental drivers, sampling strategies, and network construction methods classes of heterogeneity that can strongly influence the topology of species interaction networks created by different sets of researchers. Here, "strict" refers to the high likelihood that the data for these non-ecological systems were recorded consistently using such definitions that their respective nodes/edges would more accurately and precisely reflect their intended purpose when implemented as a network, as compared to species interaction networks. Furthermore, we either built each non-ecological network ourselves (i.e., sports networks), or used those previously built by us (i.e., all non-ecological networks other than sports were obtained from Michalska-Smith and Allesina (2019) [13]), thus ensuring appropriate data were used to build each non-ecological network. Indeed, the data used to build these networks came from specific databases for each domain (or subgroup within each domain, where subgroup refers to the different categories of a domain that networks represented, see Table 2). Moreover, as the data for each domain/subgroup of non-ecological networks came from the same database, if any class of heterogeneity were to influence their topology, the resulting heterogeneity would at least be consistent, and thereby reduce potential dispersion in measured topological heterogeneity. While undoubtedly the 3 classes of heterogeneity still influence non-ecological networks, for example, due to the misidentification of nodes, we expected that these classes would be significantly less influential than those within species interaction networks. In particular, we expected large amounts of topological heterogeneity in available species interaction networks created by different sets of researchers resulted from the inconsistent ways ecological communities were translated into networks by the different sets of researchers. We expected this would have introduced inconsistent topology across species interaction networks thereby increasing the dispersion in measured topological heterogeneity.

To avoid extremely small bipartite networks that may bias our results [13], we only included networks that had at least 5 nodes in either disjoint sets of nodes, e.g., we required at least 5 pollinator and 5 plant species in a plant–pollinator network. Additionally, only the giant component of each network was used (i.e., the largest connected component of a graph), given that it is unclear how to appropriately analyze disconnected networks.

## Directed graphlet correlation distance (DGCD)

In ecology, the most adopted subgraph technique is based on motifs. Generally, for a graph $G$ composed of a set of nodes $V$ and a set of links $L$, denoted as $G(V,L)$; a motif of $G$ is a subgraph $G'(V',L')$ with a subset of nodes $V'$ from $V$ where any edges linking the nodes of $V'$ found in $V$ are contained in $L'$ [49,50]. As differences in network structure are measured by which motifs are under-/overrepresented in the real network compared to a chosen network null model [46,51], like many statistical analyses, the results from the motif analysis depend on the choice of null model. As a consequence, motifs have been cited for possibly relying on ill-posed null models as a basis for significance testing [52].

To overcome the null model limitation, we instead adopted the subgraph technique of directed graphlet correlation distance (DGCD) [53] to characterize the topological differences between networks. Generally, DGCD evaluates network pairwise dissimilarity without relying on a network null model and does so by quantifying differences in the associations between

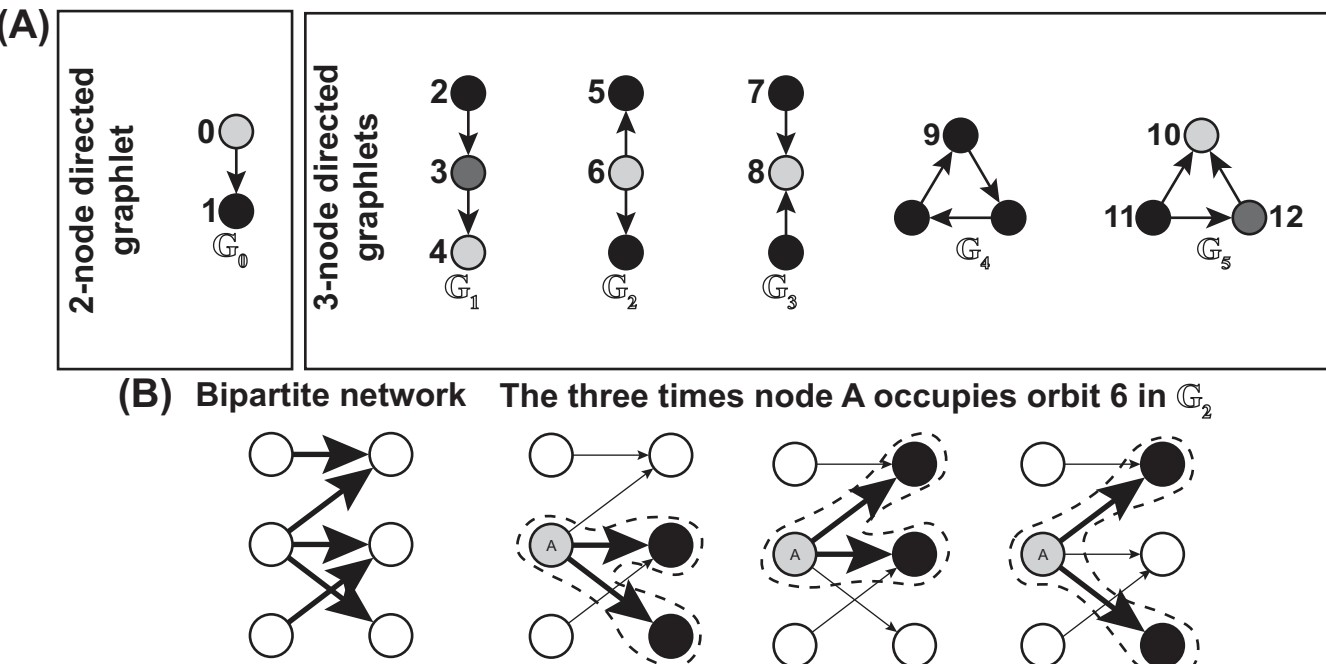

**Fig 3. (A)** The 6 directed graphlets ($G_i$) consisting of 2 to 3 nodes and their respective orbits (i.e., the corresponding 13 numerically labeled node positions). Each unique shade in a single graphlet corresponds to a unique orbit in that graphlet. Note that for the directed bipartite networks used in this study, only graphlets $G_0$, $G_2$ and $G_3$ appear. **(B)** An example calculation of the number of times node A of a directed bipartite network occupies orbit 6, where dashed lines indicate the location of $G_2$.

the appearance of directed graphlets (Fig 3A) within a given network to those of another network.

Formally, graphlets are the induced subgraphs $G'(V',L')$, consisting of a subset of nodes $V'$ from $V$ where all the edges linking the nodes of $V'$ found in $V$ are in the set $L'$. Within graphlets, nodes are often indistinguishable from one another. Take for example the graphlet $G_2$ in Fig 3A: in this case, both black nodes in this graphlet are indistinguishable, and thus form an automorphism orbit—simply orbit—of a graphlet. For this reason, there is only 2 orbits within $G_2$ labeled 5 and 6.

Generally, the DGCD relies on the directed graphlet correlation matrix (DGCM) of each network that contains Spearman's correlations between the number of times nodes appear as particular orbits with the number of times nodes appear as all other orbits within the given network (see Fig 3B for an example count of orbit 6 for a particular node of a bipartite network). For example, the Spearman's correlation between orbits 1 and 6 represented in a DGCM is calculated by taking the Spearman's correlation between: (i) a vector where each index corresponds to a specific node and the entry of that index would be the number of times that node appeared as orbit 1; and (ii) same as (i) except for orbit 6. Thus, when using all 13 orbits, DGCMs were symmetric 13×13 matrices containing the respective Spearman's correlations between the appearances of all 13 orbits within a network. Using the DGCMs, the pairwise DGCD was evaluated by measuring the pairwise Euclidean distances between all networks. See Eq (1) for a single pairwise DGCD measure between networks $K_i$ and $K_j$ using the 13 orbits from Fig 3A (termed DGCD-13 since it uses 13 orbits) and S1 Appendix Section S1.1 for an example derivation of the DGCD technique. We used DGCD in our study since recently Tantardini and colleagues (2019) [54] found that this method performs best at

characterizing and distinguishing between networks of different domains.

$$DGCD - 13(K_i, K_j) = \sqrt{\sum_{n=0}^{12} \sum_{m=n+1}^{12} \left(DGCM - 13_{K_i}(n, m) - DGCM - 13_{K_j}(n, m)\right)^2} \quad (1)$$

where $DGCM - 13_{K_i}(n,m)$ is the directed graphlet correlation matrix-13's value of network $K_i$ for orbits $n$ and $m$.

Since it is expected that networks from the same domain have similar topology, it is also expected that their DGCMs are similar, and consequently have small pairwise DGCD. Thus, when projected in visual space, networks from the same domain should be clustered together.

We calculated the pairwise DGCD-13 for all bipartite networks, where we assigned directions to the edges in the networks. Since bipartite networks are characterized by 2 sets of nodes where nodes belonging to the same set cannot have an edge, we assigned nodes belonging to 1 set to always represent a "to" direction and the other set of nodes to always represent a "from" direction in the directed edges. Simply put, this means that the DGCD-13 technique could recognize which nodes belonged to which set of nodes (e.g., which nodes belonged to the pollinator set of nodes and which nodes belonged to the plant set of nodes in a plant–pollinator network). According to these direction definitions imposed on the networks, only graphlets $G_0$, $G_2$, and $G_3$ could appear although all 6 graphlets and 13 orbits were used for better visualization—specifically Fig 4—but see S1 Appendix Section S1.4 for subsequent analyses using only the 6 orbits from graphlets $G_0$, $G_2$, and $G_3$, termed DGCD-6. Nevertheless, we note that the results presented in this article for DGCD-13 agree with those presented in S1 Appendix Section S1.4 using DGCD-6.

From all pairwise DGCD-13s, we measured the dispersion of network topology by calculating the mean pairwise distances between all networks of the same domain. In cases where

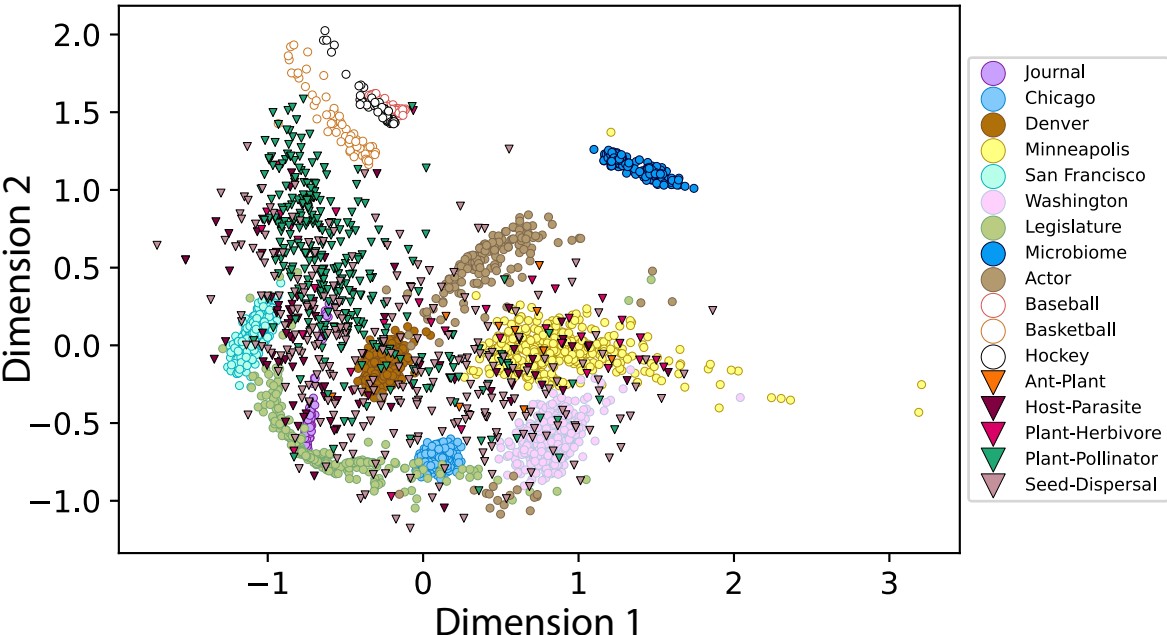

**Fig 4. Multidimensional scaling of the pairwise directed graphlet correlation distance-13 (DGCD-13) between all bipartite networks** (**n = 3,476**). Except for species interaction networks (triangles), only networks that formed clear groups in the plot are uniquely identified by color. Each point in the plot is a single network. The data and code needed to generate this figure can be found in www.osf.io/my9tv.

subgroups (e.g., hockey networks) formed coherent topology that was different from their domain (e.g., the mean pairwise DGCD-13 was much smaller for hockey networks compared to all other sports networks), we instead evaluated the mean pairwise DGCD-13 for that subgroup. If the set of networks from the same domain or subgroup had small mean pairwise DGCD-13, then this would indicate that these networks have small dispersion in their topology, i.e., they are similarly structured.

Additionally, we tested whether species interaction networks created by the same set of researchers (i.e., networks sourced from the same publication) were more topologically similar than networks not sourced from the same publication, see Table C in S1 Appendix for a list of publications that provided more than a single network. Specifically, we compared the mean pairwise DGCD-13 of networks from the same publication to the mean pairwise DGCD-13 of networks that were each a product of their own publication. Given that networks constructed by the same researchers are likely more parsimonious in terms of their topology, we expected that the mean pairwise DGCD-13 between networks from the same publication were going to be smaller than networks each produced by different publications.

## Results

The pairwise DGCD-13 between all networks was projected via multidimensional scaling (MDS) [55], also commonly known as principal coordinate analysis, using the MDS function in the Scikit-learn library [56] of Python. Except for species interaction networks, only networks that formed clear clusters were uniquely colored and identified in the MDS plot (Fig 4). Most networks from the same domain occurred in the same location in the plot and were isolated from other networks' domains in the MDS space except for species interaction, sports, and crime networks. With regards to species interaction networks, no coherent topology was observed as these networks covered all other types of non-ecological networks besides microbiome and sports networks. With regards to sports and crime networks, specific cities (i.e., the subgroups of Chicago, Denver, Minneapolis, San Francisco, and Washington) and specific sports (i.e., the subgroups of hockey, baseball, and basketball) had unique topology and formed their own respective subgroupings within the plot, and thus despite not having the same topology, there was clear topological coherence within a city's own set of crime networks and a sports' own set of networks. Here, subgrouping refers to networks from a specific subgroup that formed clear and unique clusters in the MDS plot. Since every network's domain was composed of multiple different subgroups (e.g., actor networks were made from action, adventure, ..., western movie genres/subgroups, Table 2), each domain could have potentially formed their own distinct subgroupings within Fig 4 if they exhibited unique substructure like crime and sports networks.

Of networks from the same domain or networks that had their own subgroupings within the MDS plot (Fig 4), species interaction networks had the largest mean pairwise DGCD-13 of 1.101—about twice as much as the set of legislation networks which was the next domain or subgrouping with the most topological dispersion (Table 3). This pattern also held when using median pairwise DGCD-13 (Table B in S1 Appendix) and so mean DGCD-13 was not significantly influenced by outliers. As well, the large variability in the size of species interaction networks did not contribute to this larger mean pairwise DGCD-13 value (Table D in S1 Appendix). Interestingly, both legislation and Minneapolis crime networks also had relatively high mean pairwise DGCD-13 (0.578 and 0.509, respectively), although legislation networks were composed of 4 subgroups that did not form subgroupings in the MDS plot (i.e., US House, US Senate, UN General Assembly, and European Parliament) which likely contributed to this larger value.

**Table 3. Mean pairwise directed graphlet correlation distance-13 (DGCD-13) between bipartite networks from the same domain or subgrouping.** Subgrouping refers to a subgroup [i.e., networks classified as the same type of network from the same domain during network construction (e.g., the Chicago networks in the crime network domain)] that formed an obvious cluster within the MDS plot (Fig 4). See Table 2 for a list of network domains and their corresponding subgroups. All species interaction networks were classified into their appropriate subgroup even though they did not form subgroupings.

| Network domain | Subgrouping or subgroup | Mean pairwise DGCD-13 | Number of networks |
|---|---|---|---|
| Species interaction | Ant–plant | 0.794 | 10 |
| | Host–parasite | 1.064 | 97 |
| | Plant–herbivore | 1.264 | 41 |
| | Plant–pollinator | 0.890 | 298 |
| | Seed–dispersal | 1.092 | 277 |
| | None[1] | 1.101 | 723 |
| Actor | | 0.414 | 172 |
| Crime | Chicago | 0.129 | 366 |
| | Denver | 0.180 | 366 |
| | Minneapolis | 0.509 | 366 |
| | San Francisco | 0.169 | 366 |
| | Washington | 0.305 | 366 |
| Journal | | 0.234 | 109 |
| Legislation | | 0.578 | 245 |
| Microbiome | | 0.241 | 203 |
| Sports | Baseball | 0.200 | 71 |
| | Basketball | 0.459 | 68 |
| | Hockey | 0.330 | 55 |

[1]DGCD-13 between all species interaction networks.

Exclusively within the species interaction domain, networks from the same publication were more topologically similar, by about a factor of 2, than networks that were each a product of their own publication (0.544 and 1.134 mean pairwise DGCD-13, respectively, Table 4). This smaller dispersion within networks from the same publication was also about 32% less than the topological dispersion within the ecological subgroup that had the least topological dispersion, i.e., ant–plant (0.794 pairwise DGCD-13, Table 3). It should be noted, however, that while ant–plant networks were the least topologically heterogeneous subgroup of species interaction networks tested, this should not be generalized given that we only had a few networks available to us ($n = 10$), which were sourced from only 3 publications. Nevertheless, although networks from the same publication were generally of the same species interaction subgroup (i.e., most networks from a specific publication belonged to only one of either ant–plant, host–parasite, plant–herbivore, plant–pollinator, or seed–dispersal subgroup), networks

**Table 4. Mean pairwise directed graphlet correlation distance-13 (DGCD-13) of bipartite species interaction networks from the same publication grouping.**

| Publication grouping | Mean pairwise DGCD-13 | Number of networks | Number of publications |
|---|---|---|---|
| One network per publication | 1.134 | 236 | 236 |
| Multiple networks per publication | 0.544[1] | 487 | 58 |

[1]Calculated by taking the mean of the average pairwise DGCD-13s between networks from the same publication, weighted by the number of networks produced by each publication.

Multiple bipartite networks sourced from the same publication (i.e., networks created by the same set of researchers) are termed "multiple networks per publication" and bipartite networks sourced from publications that each produced only a single network are termed "one network per publication." See Table C in S1 Appendix for a list of publications that provided more than 1 network and each publication's mean pairwise DGCD-13.

from the same publication were more topologically similar than any single species interaction subgroup.

## Discussion

Ecologists commonly reuse species interaction networks created by different sets of researchers to test how ecological and environmental processes shape network topology across space and time [8,9,26]. However, unwanted topological differences as a result of the different ways in which researchers translate ecological communities into networks could inhibit their commensurability [10,23,24]. When assessing the degree of topological heterogeneity, i.e., the total amount of topological differences between a group of networks, we find that species interaction networks created by different sets of researchers are extremely topologically heterogeneous—about twice the amount than the next most heterogeneous network domain tested—and that this large heterogeneity is linked to the publication source of each network. Altogether, these findings suggest that species interaction networks created by different sets of researchers can be problematic for deducing ecological topological rules since much of the topological heterogeneity is likely not due to ecological processes as is often assumed.

A general principle in statistics is that an increased sample size reduces uncertainties of estimators [57]. Armed with this principle, and the ease with which species interaction networks can be obtained from online resources [25], it may then be tempting to assume that increasing one's data set by collecting all possible networks available alleviates any data issues. However, using the largest set of bipartite species interaction networks available ($n = 723$), we illustrate how large amounts of topological heterogeneity (via the mean pairwise DGCD-13, Table 3) and consequently uncertainty exists in the topology of species interaction networks created by different sets of researchers, confirming that more data is not always better when biases are present [57]. While some metrics, including sampling intensity and effort, have previously been used to control for biases and sources of topological heterogeneity in species interaction networks [43], these controls do not effectively account for all sources of heterogeneity (e.g., differences in node taxonomy across networks) or when using different topological metrics (e.g., modularity, nestedness). Hence, careful consideration, beyond a single metric of control, is required when deciding which networks to include in one's analyses, so that the majority of topological differences measured between species interaction networks are a result of the ecological process-of-interest and not from confounding factors.

The large amount of topological heterogeneity in species interaction networks created by different sets of researchers likely reflects their topological uniqueness due to the distinct (a)biotic conditions each represented community experiences, the distinct sampling strategies adopted to characterize each ecological system as a network, and the distinct construction methods used to create each network (Table 1). Indeed, the large difference in the amount of topological heterogeneity between species interaction networks and non-ecological networks may be attributed to these 3 classes of topological heterogeneity given that the non-ecological networks were created in a consistent way to try to eliminate much of their influence. For example, we built non-ecological networks using data attained from consistent sampling strategies (e.g., each sampled crime network represented a specific city and day of the year in 2016) and we used consistent network construction definitions when building the networks from the data (e.g., all interactions in crime networks always represented a type of crime occurring in a city's specific neighborhood). This is not to say that non-ecological networks are devoid of their own sources of topological heterogeneity. For example, differences in both voter sentiment across time and differences in the political landscape across space within the legislative networks (e.g., between US House and UN General Assembly networks) likely contribute

some topological heterogeneity. However, the biological and environmental drivers, sampling strategies, and network construction methods classes of heterogeneity seem to be accentuated in species interaction networks created by different sets of researchers as compared to the tested non-ecological networks.

Importantly, even though biological and environmental drivers, sampling strategies, and network construction methods classes of heterogeneity are known to influence the topology of species interaction networks, they are nevertheless rarely acknowledged or appropriately controlled in ecological studies. This is especially problematic when reusing networks created by different sets of researchers since the influence of these classes are likely to vary considerably depending on the methods and approaches that different researchers use to create each network. In fact, rarely are differences in sampling strategies controlled for when reusing networks, even though sampling strategies influence network topology. For example, species interaction networks are already topologically different when constructed from observational data collected over different amounts of time [18–20], or over different amounts of area [35,36]. Furthermore, related to variations in sampling strategies, networks may also vary in their sampling sufficiency [58]. Insufficiency can occur when the sampling design does not match the biology of the community and can make networks incommensurable even when networks are built using the same sampling strategies. Moreover, differences in biological and environmental drivers that ecological communities experience are sometimes not controlled for when reusing networks, even though these drivers can influence network topology. For example, species interaction networks are already topologically different depending on the temperature each community experiences [31]. As well, despite the widespread reuse of species interaction networks created by different sets of researchers for testing ecological hypotheses, it is still relatively unknown how different network construction methods influence topology, which may also make network comparison difficult. For example, interactions in one plant–pollinator network can represent a pollinator touching a plant and in another network represent pollen of a plant being found on a pollinator [59], or networks may or may not contain pollinators which are commensals or parasitic to plants [60]. Thus, without care and appropriate control of the topological differences from the 3 heterogeneity classes, one is liable to find erroneous relationships when reusing species interaction networks created by different sets of researchers [45,61].

All is not lost when reusing species interaction networks created by different sets of researchers, as one approach to avoid a large amount of the topological heterogeneity may be to attempt to control for the publication source of each network. While we found a large amount of topological heterogeneity between all species interaction networks, we also found that networks created by the same set of researchers (i.e., networks from the same publication) were more topologically similar to each other (Table 4). Interestingly, we also found that networks from the same publication were more topologically similar than networks from any species interaction subgroup (i.e., networks belonging only to either ant–plant, host–parasite, plant–herbivore, plant–pollinator, or seed–dispersal). Consequently, it appears that publication has an even greater impact on the topology of species interaction networks than biological processes alone. This may occur since researchers of a given publication generally construct networks under parsimonious conditions [10], for example, by observing and characterizing ecological communities across the same time duration (e.g., Trøjelsgaard and colleagues (2015) [62]) or by classifying nodes across networks using the same protocol (e.g., Pereira Martins and colleagues (2020) [63]), and thus inadvertently control for many sources of topological heterogeneity. Of course, biological effects are likely influencing the topology of all networks but they can be more easily obscured when analyzing networks across publications. It is then likely that controlling for the effect of publication can reduce unwanted topological

heterogeneity between networks. We do, however, strongly caution those that only attempt to account for the publication source of each reused network. Similar to how different network metrics are sensitive to different amounts of sampling sufficiency [58], the strong similarity between networks from the same publication may be more or less relevant when investigating network structure using other metrics.

Although most researchers do not originally intend for their networks to be reused and compared to other networks, often they are included in meta-analysis type studies if they are made freely available. Original authors of networks can improve the scientific utility of their networks by providing other researchers with information about how they were constructed [26]. In particular, by providing detailed network metadata, including information on relevant biological and environmental drivers, sampling strategies, and network construction methods, authors of the networks can help others understand the specific conditions under which each network was created. Additionally, given the recent developments of composite methods designed to estimate sampling sufficiency for ecological networks (e.g., Casas and colleagues (2018) [58]), authors of species interaction networks could also calculate this metric or provide the information to do so to check if communities have been sufficiently sampled. Then, beyond controlling for sources of topological heterogeneity (e.g., node taxonomy), researchers reusing these networks could also control for sampling sufficiency which is another means to improve network commensurability. Given appropriate metadata, researchers could also study how each class of heterogeneity influences the topology of species interaction networks, rather than the totality of topological heterogeneity as we have done here.

Nevertheless, as users of species interaction networks that happen to be constructed by different sets of researchers, the onus is on us to know the limitations of our data and to ensure that they effectively represent the systems in the corresponding models we use [64]. Given that all species interaction networks are models and are thus subject to imperfections (e.g., Pringle and Hutchinson (2020) [30]; Thomson (2021) [65]), we should be aware of their overall shortcomings and attempt to correct for them, especially since our findings are often used to inform policy aimed at conserving ecological systems.

## Caveats

A limitation in our analyses was the use of small species interaction networks (e.g., $<100$ nodes). Since networks with a small number of nodes and edges are generally more difficult to classify than larger networks [46], we perhaps inadvertently increased the perceived topological heterogeneity of species interaction networks as compared to some of the non-ecological networks. Regardless, the crime networks we used were of similar size to species interaction networks (Table A in S1 Appendix), but were less topologically heterogeneous (Table 3 and Fig 4). Clearly then, it was still possible to find topological consistency even in small networks, but less so when networks were both small and ecological. This suggests that the topological heterogeneity in species interaction networks created by different sets of researchers was due to more than just the difficulty of classifying small networks, but likely also from biological and environmental drivers, sampling strategies, and network construction methods classes of heterogeneity, which reused networks created by different sets of researchers are especially prone to. Importantly, this same problem of using small networks is also relevant when applying any other types of metrics to ecological networks, e.g., nestedness and modularity.

Although we generally failed to find pervasive and coherent topology within species interaction networks created by different sets of researchers, we highlight that our results do not necessarily invalidate patterns others have found (e.g., high nestedness in plant–animal networks

[66]). Instead, these patterns are perhaps true under strict conditions, such as controlling for the unwanted differences in topology between studies when reusing their networks.

## Conclusion

Species interaction networks created by different sets of researchers likely suffer from comparison problems due to many sources of topological heterogeneity, i.e., via biological and environmental drivers, sampling strategies, or network construction methods classes of heterogeneity. Quantitatively, our findings show that these species interaction networks are remarkably topologically diverse and that we should be especially careful when reusing this source of data for deducing rules of community assembly, perhaps by controlling for the publication source of each network.

## Supporting information

**S1 Appendix. Detailed and additional methods, supplementary figures, and tables.**
(PDF)

## Acknowledgments

We thank all the authors of previous publications that made their network data freely accessible. Without their contributions to Open Science, this study would not have been possible.

## Author Contributions

**Conceptualization:** Chris Brimacombe, Korryn Bodner, Matthew Michalska-Smith, Timothée Poisot, Marie-Josée Fortin.

**Data curation:** Chris Brimacombe.

**Formal analysis:** Chris Brimacombe, Korryn Bodner.

**Investigation:** Chris Brimacombe, Korryn Bodner, Marie-Josée Fortin.

**Methodology:** Chris Brimacombe, Korryn Bodner, Matthew Michalska-Smith, Marie-Josée Fortin.

**Supervision:** Timothée Poisot, Marie-Josée Fortin.

**Validation:** Chris Brimacombe, Matthew Michalska-Smith.

**Visualization:** Chris Brimacombe.

**Writing – original draft:** Chris Brimacombe, Korryn Bodner.

**Writing – review & editing:** Chris Brimacombe, Korryn Bodner, Matthew Michalska-Smith, Timothée Poisot, Marie-Josée Fortin.

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
