## [Editor Report · Decision Letter 0]

15 Aug 2022

Dear Dr Brimacombe, 

Thank you for submitting your manuscript entitled "Shortcomings of reusing species interaction networks created by different sets of researchers" for consideration as a Research Article by PLOS Biology.

Your manuscript has now been evaluated by the PLOS Biology editorial staff, as well as by an academic editor with relevant expertise, and I'm writing to let you know that we would like to send your submission out for external peer review.

IMPORTANT: We think that your study would be better considered as a "Meta-Research Article." Please change the article type to "Meta-Research Article" when you upload your additional metadata (see next paragraph).

Once your full submission is complete, your paper will undergo a series of checks in preparation for peer review. After your manuscript has passed the checks it will be sent out for review. To provide the metadata for your submission, please Login to Editorial Manager (https://www.editorialmanager.com/pbiology) within two working days, i.e. by Aug 17 2022 11:59PM.

Kind regards,

Roli Roberts

Roland Roberts, PhD

Senior Editor

PLOS Biology

rroberts@plos.org

---

## [Decision Letter · Decision Letter 1]

2 Nov 2022

Dear Dr Brimacombe,

Thank you for your patience while your manuscript "Shortcomings of reusing species interaction networks created by different sets of researchers" was peer-reviewed at PLOS Biology. It has now been evaluated by the PLOS Biology editors, an Academic Editor with relevant expertise, and by four independent reviewers. 

The reviewers' comments are broadly positive, but two of them (reviewers #3 and #4) raise some significant concerns that must be addressed. You’ll see that reviewer #1 is positive, but wants a new Figure to illustrate DGCD, and some discussion of the implications of high DGCD values, and some textual clarifications. Reviewer #2 is enthusiastic and has no requests. Reviewer #3 is concerned that the you may not be accounting for network size, asks about the behaviour of the DGCD-13 metric, and suggests a number of statistical tests. Reviewer #4 is positive, but has similar concerns to rev #3 about the validity of your choice of non-ecological networks; s/he also thinks that the you need to seriously consider sampling sufficiency.

In light of the reviews, which you will find at the end of this email, we would like to invite you to revise the work to thoroughly address the reviewers' reports.

Given the extent of revision needed, we cannot make a decision about publication until we have seen the revised manuscript and your response to the reviewers' comments. Your revised manuscript is likely to be sent for further evaluation by all or a subset of the reviewers.

**IMPORTANT - SUBMITTING YOUR REVISION**

*Re-submission Checklist*

*Published Peer Review*

*PLOS Data Policy*

*Blot and Gel Data Policy*

Sincerely,

Roli Roberts

Roland Roberts, PhD

Senior Editor

PLOS Biology

rroberts@plos.org

REVIEWERS' COMMENTS:

Reviewer #1:

I have read the manuscript on the meta-study performed to highlight the importance of considering the publication source when reusing and analyzing species interaction networks. It think this is the main message of this manuscript. It is already known that the species interaction networks available in databases, like the Web of Life database, are highly heterogenous and that a careful consideration should be taken when comparing network topology across networks. The authors quantified that heterogeneity with a novel metric and compare the similarity of species interaction networks with the similarity of non-biological networks that were built using a much more standardized approach. That is novel and interesting. In general, I like the paper and I think it meets the requirements for its publication. However, some methodological questions on the Directed Graphlet Correlation Distance (DGCD) should be clarified. Specifically, a new figure illustrating how those graphlets are embedded within a bipartite network (replacing current Figure 1). I also miss a discussion on to what extent high values of DGCD might or might not invalidate the existence of specific network patterns, like nestedness and modularity, that have been found in many species interaction networks.

Comments to Authors:

I think the authors have done a great job in this meta-study. I like the motivation of this manuscript, the novel approach they have followed to compare network topology, and the extent of the dataset they have used. Yet, I miss an explicit interpretation in the discussion section on how the global metric applied (DGCD) might or might not invalidate the existence of specific network patterns that have been widely detected across species interaction networks (e.g., nestedness and modularity). Some methodological questions should also be clarified. Below, I have listed some minor comments that I think will improve the reader's understanding.

On Methods:

- I think current figure 1 does not clearly depict the idea underlying the directed graphlet correlation distance. Perhaps, a new one representing a couple of graphlets embedded within a small bipartite network would be much more informative.

- I would appreciate a clarification on the distinction between graphlets and network motifs.

- Why did you measure the dispersion of network topology by calculating the average pairwise distances between all networks of the same domain? Wouldn't be the variance a much more informative descriptor?

On Results:

- I do not understand how you go from the pairwise DGCD-13 between all networks and the multidimensional scaling represented in figure 4. I might misss that step, but you only provide a single sentence at the begining of the results section: "The pairwise DGCD-13 between all networks was projected via multidimensional scaling" (line 154).

- Networks published in the same publication were built usually in different places on the same time scale using a standardized method to answer a specific question, and in most cases, for a single type of interaction. Therefore, we expect them to be more similar topologically than networks that were compiled from different authors with different motivations. For this reason results must be interpreted with caution: "networks from the same publication were more topologically similar, and thus publication had an even greater effect on the topology of networks than biology alone" (lines 182-183).

On Discussion:

- Species interaction networks found in open databases are highly heterogeneous and we should pay more attention to the particularities of each network before performing any global analysis. I totally agree. Specifically, a careful consideration should be taken when comparing species interaction networks. But we could also see the glass as half-full and say that, even with the large heterogeneity in biotic and abiotic conditions, temporal and spatial scales, sampling effort and methodological approaches, there are some network properties that seem to pervade species interactions networks. DGCD is a global measure of the topological differences among networks, but I think that metric does not invalidate the existence of the same specific pattern that might be hidden under networks with different DGCD (e.g. nestedness). In fact, the suggestion you provide on controlling for the publication source of each network as a fruitful approach when reusing and analyzing species interaction networks is a powerful message (lines 261-263).

Kind regards,

Miguel A. Fortuna.

Reviewer #2:

[identifies himself as Christian Mulder]

Dear Authors,

My most sincere compliments for your outstanding revision and for the challenging topic chosen for your research. It is an extremely important issue in Food Web Ecology and I warmly welcome your work, very well done!

Kind regards, Christian Mulder

Reviewer #3:

[identifies himself as Adrian David González Chaves]

I think is a relevant tropic and enjoy the fluidity while reading the work; the authors present an issue which is currently relevant and meant to be more so with time. Meta-analysis and synthesis analysis are become popular and needed and cautions about how to proceed with them are crucial (Mestre et al. 2022; Xing & Fayle 2021). Nonetheless, I feel that the main issue that the authors claim to tackle is not being fully address and could be explored with the data available, as they use a very extensive database. The following paragraphs are intended to rise the concern about how the authors explores the variability between networks and try to exemplify how could the fully answer the questions raised in the introduction.

Authors main claim is that variability between network structure of ecological interaction is greater than other networks, e.g. social networks and microbial networks. Clearly watching from the supplementary material, the sizes of the ecological networks are inferior to any of the other networks considered (Table A1). Knowing that network size is fully influential on networks structure I wonder if this somehow accounted in the DGCD-13 metric considered. Furthermore, the none-ecological networks seem to have a lot of standardization, less prone to the topological heterogeneity the authors expect to fin in ecological networks. While only for Crime and Sport networks there are subgroupings (states and sport discipline) for the rest of the networks there are not categories, while for ecological networks there are either seasonal (already explored by the first author in another work), latitudinal mentioned in the literature cited, networks might have been gathered through different sampling method. Thus, it is expected that such variability will be found. The novelty would be whether the DGCD-13 methodology is good for assessing how much of the variability can be attributed to each topological heterogeneity class and not as a response of unwanted topological distances

I don't about this metric, but I have some basic questions relating this metric. Is it possible to build a probability test over the values it generates? How do you known that the 76% of difference in variation reported doesn't occur by chance or if it is influenced for some extreme values within the 723 networks considered. For instance, and regarding the former comment, the size variability of the ecological networks considered are higher when compared to the other networks, so I strongly suggest incorporating a test that allow to consider such factor of variability, given this is present both for ecological and non-ecological networks. You consider the mean values to compare, but how much of the mean variability values are a result of the variance in network size. From what I understood the DGCD-13 analysis considers graphlets (or motifs), which are known to be influenced by the networks size, as it occurs with the networks structure (Milo et al. 2002 - Science: DOI: 10.1126/science.298.5594.824). 

In the abstract you mentioned: "to evaluate the degree to which this topological heterogeneity is present in available ecological networks, …". I feel this is not fully explored from your data and could be explored with a statistical test. I wonder whether you could use the mean DGCD-13 value as a starting comparison point, to compare how much each network deviates from it, and then test how much of each heterogeneity class could explain the variability found. Then you would have an attributable value to how much of the biological, sampling or network construction is predicting the network structure variability. The work you refer to, where you used the same data, does this (Brimacombe et al 2022) using seasons, but in that case, you are not looking to network structure variability as you calculated here. Meaning you have information on network size and temporal elements of the studies to assess how much of the higher variability found can be attributed to each topological feature and address what is mentioned in rows 39 to 41. Because from what I understood, the quantification that you say is going to be address in rows 67 to 70, is not really undertaken. You seem to infer the role of the topological heterogeneity from comparing variability among researchers and between ecological networks and other networks, and not through an integrated analytical approach. 

Probably why your discussing and conclusion are seems to be calling attention to considering this higher variability when implementing meta-analysis, but for me it suggests that give the existence of the high variability there is space for testing how much other factors (or grouping) explain the variability found (as you seem to have done for the other referenced research). Again, could you not create a model to estimate how much of the variation found in the Ecological networks can be attributed to each of the topological categories mentioned? How is that you can disentangled between each category? To illustrate the need to take this into account, and how to do so.

Reviewer #4:

I have read the manuscript PBIOLOGY-D-22-01700R1 by Brimacombe et al. with much interest. The authors evaluate the topological heterogeneity of ecological networks, comparing ecological networks to "non ecological" networks, and by assessing the heterogeneity of networks from ecological studies that have one network vs studies that have multiple networks. The authors found that the topology of ecological networks are much more heterogenous than "non-ecological "networks, and that interaction networks from the same publication are much more topologically similar to each other than interaction networks that come from the same study. They suggest that three different sources are likely to be the main causes of the heterogeneity they found in ecological networks, namely: sample strategy, bio-ecological drivers, and how networks are constructed. The question tackled by the authors is very interesting and important, given that in the past years, ecologists have tried to produce syntheses and tried to understand biogeographical patterns in species interaction networks amassing data from distinct publications, ignoring possible flaws with the available data. The authors also suggest some possibilities to deal with this problem. The results are interesting, and I believe the manuscript is an important contribution to the field of ecological networks. Nonetheless, I do have some concerns, that I think it should be addressed and some minor requests for change and/or suggestions, that hopefully should help to improve the manuscript and not take much time. 

Although I appreciate the efforts of the authors to compile and amass this amount of data, I was not convinced of some of the assumptions regarding the "non ecological networks", that make them a significant baseline to estimate the topological heterogeneity of ecological networks. The authors claim that ecological networks are hampered by biological, environmental drivers, sampling strategy, and/or network construction, and that the "non ecological" networks are not hampered by these sources of heterogeneity. However, the authors do not provide any evidence of references that can support this fundamental claim of the manuscript. Although, I do not have much familiarity with these kind of data, I found it hard to believe that these data sets don't have sources of biases or are not hampered by problems related to sampling sufficiency. For instance, crime reports are known to be biased, and I guess this could have an impact, I imagine that both author and actor networks will depend on the database being used, plus, they might share the same problem with ecological networks in regard to the misidentification of nodes, due to homonymous authors/actors, etc. I do not understand how microbiome networks were considered as "non"- ecological networks. Why are they not hampered by biological or environmental drivers, and how they can serve as the basis for you comparison here? Most importantly, microbiome sampling strategy also varies a lot as "ecological" networks do; e.g., gut microbiome networks can be constructed based on samples collected from feces, mucosal biopsy, intestinal fluid, etc. or come from different parts of the gastrointestinal tract.

I also wonder, even if those assumptions related to the "non-ecological" networks hold, if the variability found here could not be due to the fact that bipartite networks are more variable than unipartite networks, which form your entire reference system? Is there any evidence that can support that?

Most importantly, I think one of key aspect missing here is the problem related to sampling sufficiency/completeness, which is a fundamental aspect of sampling theory that is largely overlooked by network ecologists. There is no way to move forward, unless network ecologists start to pay attention and report that. Although this has been a pivotal problem in both population and community ecology, network ecologists seem to get a free pass, not accounting for it or using inappropriate methods to estimate sampling suffciency. However, most networks published to date are likely to be under-sampled, and to make things more complicated, evidence suggest that sampling sufficiency is metric specific, even for the same dataset (Casas et al. 2018. Assessing sampling sufficiency of network metrics using bootstrap. Ecological Complexity 36 268-275), which means that, even if networks have been properly sampled in order to estimate a specific topological property, they may not be commensurable, if original studies were designed to estimate different topological properties. 

There has been some great efforts in the past years in order to create appropriate methods to estimate sampling sufficiency for ecological networks, see for instance: Casas et al. (2018), Jordano (2016. Sampling networks of ecological interactions. Funct Ecol, 30: 1883-1893) and references therein.

At last, different from community ecology, recording pairwise species interaction demands that network ecologists account for species-specifics detection probability, and link(interaction)-specific detection probability. I wonder why the authors did not consider analytical solutions, such as hierarchical modeling as a possible solution form this problem, since it can account for detection probabilities, for nodes and link and account species misidentification. I like the direction given by the authors in terms of using studies as some sort of random effect, however the solution presented here will not suffice for the problems exposed in the manuscript, if authors of original studies do not account for sampling sufficiency and problems relate to node/link detection rates

Minor comments:

L71-72. I would suggest rephrasing this sentence; as it gave me the impression that the authors were going to partition the heterogeneity among these three sources, which is not the case here. I think other readers would build the same expectation.

L121. The word "best" is duplicated in this sentence.

L199-202. That is assuming that networks published in each study has been properly samples and that topological estimators have reached some asymptotic property, which is unlikely. The problem here is that poorly, under- sampled networks are published, and then compiled together in syntheses/biogeography studies.

L154: What is the exact method used for the MDS? Principal Coordinates?

L174. What is the average and/or range of the number of networks in publications with multiple networks? Could be that the small number of networks within a publication cause this small variability for the "multiple networks per publication grouping? Once again poor sampling sufficiency for this multiple networks could also be the factor influencing the results here. Is there any evidence that sampling sufficiency change between these two groupings for a specific metric?

L199-210. Once again, assuming that each and every network has reached sampling sufficiency, for the problem at hand. See comments above on metric specific sampling sufficiency.

L 259. Standardizing sampling time, does not guarantee that sampling sufficiency is reached given that detection rates are note homogeneously distribute across space… Species specific detection rates and iteration specific detection rates can between habitats.

L267: Maybe most importantly here, is providing sampling sufficiency for the parameters that were investigated. 

Fig 1 and table 1 should or could also depict the fact that species specific detection probabilities, and interaction detection probabilities vary.. so even if a researcher have the same sampling strategy across space and time, may not be a sufficient strategy to deal with the problem.

---

## [Decision Letter · Decision Letter 2]

22 Feb 2023

Dear Dr Brimacombe,

Thank you for your patience while we considered your revised manuscript "Shortcomings of reusing species interaction networks created by different sets of researchers" for publication as a Meta-Research Article at PLOS Biology. This revised version of your manuscript has been evaluated by the PLOS Biology editors, the Academic Editor, and three of the original reviewers.

Based on the reviews, we are likely to accept this manuscript for publication, provided you satisfactorily address the remaining points raised by the reviewers and the following data and other policy-related requests:

IMPORTANT: Please attend to the following:

a) Please address the remaining requests from reviewer #1.

b) Please provide a blurb, according to the instructions in the submission form.

c) Many thanks for providing the underlying data and code in OSF. Please cite the location of the data clearly in all relevant main and supplementary Figure legends (probably Figs 4, S5, S6 and S7?), e.g. “The data and code needed to generate this Figure can be found in https://osf.io/my9tv/"

We expect to receive your revised manuscript within two weeks. 

*Published Peer Review History*

*Press*

Sincerely,

Roli Roberts

Roland Roberts, PhD

Senior Editor,

rroberts@plos.org,

PLOS Biology

DATA NOT SHOWN?

REVIEWERS' COMMENTS:

Reviewer #1:

The main points raised in my previous review have been satisfactorily addressed. I just have a couple of very minor comments:

1) I am wondering why you computed DGCD-13 instead of DGCD-6 (I am aware you reported DGCD-6 as well in the Appendix). You mention in Fig.3 that "Note that for the directed bipartite networks used in this study, only graphlets G0, G2, and G3 appear". Then, a node can occupy only 6 orbital positions (0, 1, 5, 6, 7, and 8). Why did you use 13 orbits to compute DGCD? You seem to justify this decision by saying: "According to these direction definitions imposed on the networks, only graphlets G0, G2, and G3 could appear although all six graphlets and 13 orbits were used for better visualization" (lines 184-185). I would appreciate a much informative explanation.

2) Fig. S2: the number of times node A occurs at orbital position "0" is 3, but I think it should be 1, right?

Thanks for your thoughtful revision.

Kind regards,

Miguel A. Fortuna

Reviewer #3:

[identifies himself as Adrian David González Chaves]

Thank you for addressing all the comments. I'm pleased with your edits and enjoyed reviewing this manuscript. I learned something new in the process.

Reviewer #4:

I thank the authors for considering my comments and suggestions in this revised version of their manuscript and for producing such a detailed and nice rebuttal letter. Thank you for the work!

---

## [Editor Report · Decision Letter 3]

7 Mar 2023

Dear Dr Brimacombe,

Thank you for the submission of your revised Meta-Research Article "Shortcomings of reusing species interaction networks created by different sets of researchers" for publication in PLOS Biology. On behalf of my colleagues and the Academic Editor, Pedro Jordano, I'm pleased to say that we can in principle accept your manuscript for publication, provided you address any remaining formatting and reporting issues. These will be detailed in an email you should receive within 2-3 business days from our colleagues in the journal operations team; no action is required from you until then. Please note that we will not be able to formally accept your manuscript and schedule it for publication until you have completed any requested changes.

Sincerely,

Roli Roberts

Senior Editor

PLOS Biology

rroberts@plos.org